# Feasibility of Using the Hexoskin Smart Garment for Natural Environment Observation of Respiration Topography

**DOI:** 10.3390/ijerph18137012

**Published:** 2021-06-30

**Authors:** Shehan Jayasekera, Edward Hensel, Risa Robinson

**Affiliations:** Department of Mechanical Engineering, Rochester Institute of Technology, Rochester, NY 14623, USA; gbj6142@rit.edu (S.J.); echeme@rit.edu (E.H.)

**Keywords:** respiration topography, waterpipe, hookah, combustible cigarettes, wearable respiratory monitor, lung volume, inhalation topography

## Abstract

*Background*: Limited research has been done to measure ambulatory respiratory behavior, in particular those associated with tobacco use, in the natural environment due to a lack of monitoring techniques. Respiratory topography parameters provide useful information for modeling particle deposition in the lung and assessing exposure risk and health effects associated with tobacco use. Commercially available Wearable Respiratory Monitors (WRM), such as the Hexoskin Smart Garment, have embedded sensors that measure chest motion and may be adapted for measuring ambulatory lung volume. *Methods*: Self-reported “everyday” and “some days” Hookah and Cigarette smokers were recruited for a 3-day natural environment observation study. Participants wore the Hexoskin shirt while using their preferred tobacco product. The shirt was calibrated on them prior to, during, and after the observation period. A novel method for calculating the calibration parameters is presented. Results: N_H_ = 5 Hookah and N_C_ = 3 Cigarette participants were enrolled. Calibration parameters were obtained and applied to the observed chest motion waveform from each participant to obtain their lung volume waveform. Respiratory topography parameters were derived from the lung volume waveform. *Conclusion*: The feasibility of using the Hexoskin for measuring ambulatory respiratory topography parameters in the natural environment is demonstrated.

## 1. Introduction

Limited research has been conducted to study the respiration behavior of tobacco use in the natural environment, in part due to the lack of suitable ambulatory monitoring techniques. Current commercially available Wearable Respiratory Monitors (WRM) may be leveraged to measure ambulatory respiration topography in the natural environment [1], although little research [2] has been done to utilize these devices for this application. In this paper, we demonstrate the feasibility of using one such WRM, the Hexoskin Smart Garment (Carré Technologies Inc., Montréal, PQ, Canada), to measure ambulatory respiration topography of cigarette and hookah smokers in their natural use environment.

To fully assess the health effects associated with tobacco use, both puffing and respiration behaviors must be considered. Traditionally, researchers have focused only on puffing behavior (puffing topography) quantified by puff flow rate, puff duration, puff volume, and inter puff gap, typically measured using a topography monitor [3,4,5,6]. However, puffing topography by itself is insufficient for understanding the distribution of toxic constituents in the respiratory tract beyond the oral cavity. The flow dynamics of the inhaled and exhaled volume, characterized by the flow rate, volume, duration, start and end time of each inhale and exhale, breath-hold period, and time between breaths, are necessary for accurate modeling of deposition of inhaled tobacco particles in the lungs [7]. These parameters are collectively referred to as respiration topography.

Researchers have shown that users may alter their puffing and respiration behavior to compensate for the reduction in nicotine dose, typically by increasing puff and inhalation volume [8,9,10,11], but other compensatory mechanisms include regulating inhalation flow rate, increasing breath-hold duration, and extending the exhalation period to increase the retention of nicotine [12]. Compensatory behavior has the potential to render alternative tobacco products, such as low-yield cigarettes, ineffective at reducing the risk of exposure or adverse health effects.

Respiration topography may also be used to study the effects of product characteristics on usage behavior due to variations in flow path restriction. Research has identified two main smoking patterns: mouth-to-lung (MTL) [13] and direct-to-lung (DTL) [14,15]. MTL is typically associated with cigarette use. One possible reason for this is due to the restrictive nature of the cigarette flow path, which makes deep inhalations difficult. In contrast, we expect to see more DTL behavior in users of lower-resistance tobacco products, such as hookah and certain e-cigs, such as the JUUL. For any fixed inhaled volume, the concentration of nicotine is likely higher in DTL than in MTL since, in DTL, the volume is comprised entirely of emissions from the tobacco product, whereas in MTL, it is a smaller volume of puffed emissions that is then diluted in a larger volume of clean air during inhalation. Given this, it is conceivable that tobacco manufacturers are designing their products to have lower resistance, as is the trend from cig-a-likes, to sub-ohm, and finally to pod-style e-cigs, so as to facilitate DTL behavior and thereby increase nicotine consumption. This is, however, unverified, but with a method to measure respiratory topography, we may in the future be able to link product characteristics (e.g., flow path resistance) to product use behaviors (e.g., DTL vs. MTL) and their consequent health effects. By observing when inhalation occurs in relation to the puff, the full topographic smoking pattern can be obtained.

This paper aims to assess the feasibility of adapting the Hexoskin for ambulatory respiration topography observation and address some of the challenges, in particular, (1) calibration of the Hexoskin Smart Garment, (2) calculation of respiration topography parameters observed from smokers in their natural environment, and (3) acceptability of the Hexoskin by the participants.

## 2. Materials and Methods

### 2.1. Human Subject Study Protocol

The study protocol consisted of an online participant recruitment and pre-screening survey (Table 1) prior to the intake meeting, a final screening and verification of eligibility according to the Inclusion/Exclusion Criteria during the intake meeting, the review and signing of the informed consent form, the deployment of the WRM for a 3-day observation period with an in-lab WRM calibration on each observation day, an exit questionnaire assessing the acceptability of the monitor and the study protocol, and finally the incentives. The study protocol was reviewed and approved by the Rochester Institute of Technology (RIT) Human Subjects Research Office Institutional Review Board (IRB).

Cigarette and hookah users were recruited from the RIT community via mass email and flyers between July 2019 to March 2020, when the study was halted due to the COVID-19 pandemic. To satisfy the Inclusion Criteria, the participant must: (1) Be of legal smoking age, (2) Answer “Some days” or “Everyday” to the question Q1 or Q2 in Table 1, and (3) Indicate that they had their own tobacco product since the study would not provide them with a product to use. We complied with the change in legal smoking age from 18 to 21 on 20 December 2019 due to the “Tobacco 21” legislation [16]. The Exclusion Criteria excluded the following participants from being eligible: those with underlying cardiovascular and pulmonary diseases, pregnant women, or women intending to become pregnant.

The observation period spanned three days. On the first day, prospective participants were invited to the laboratory for a final eligibility screening and were enrolled once they reviewed and signed the informed consent form. Enrolled participants were provided with the WRM along with instructions on how to use it. The participants were informed that they must wear the WRM whenever they were using the tobacco product type they were enrolled to use and were not required to wear the WRM when not smoking. The participants were also provided with a daily study log to self-report their use behavior and compliance to the study protocol. At the end of the intake meeting, a WRM calibration was conducted on the participant, and then they were allowed to leave and use their tobacco product freely. On the second day, the participants returned to the laboratory at a pre-scheduled time for another WRM calibration. The participants were free to use their desired product prior to and after the scheduled meeting. On the third day, the participants returned to the laboratory and performed a final calibration. The participants were asked about the acceptability of the WRM and the study protocol in a structured interview. Finally, participants were provided a $25 incentive upon successful completion of the study protocol. Participants who dropped out of the study prematurely were provided with a $5 incentive.

### 2.2. Wearable Respiratory Monitor

The WRM used in this study is the Hexoskin Smart Garment with the Hexoskin Smart Device (datalogger). The Hexoskin has been used in a number of applications, including sports and fitness tracking [17,18], patient monitoring [19], and assessing obesity risk [20]. The use of the Hexoskin for ambulatory measurement of respiration topography was previously qualitatively assessed [1]. The Smart Garment is a shirt-type WRM, made from a tight form-fitting material and can be worn on its own or as an undergarment. The shirt has two embedded respiratory inductance plethysmograph (RIP) sensors, one at the thorax (right below the pectoral muscle) and one on the abdomen. These sensors measure chest motion by measuring the changes in cross-sectional area at each location on the torso. The datalogger provides power to these sensors and collects data from them at a fixed rate of 128 Hz and stores the data internally. Data collection begins immediately once the datalogger is attached to the shirt and continues until it is detached from the shirt or the datalogger runs out of battery. Each time the datalogger is attached to the shirt, a new data record is created, without overwriting previous records. At the end of the observation period, the data was extracted from the datalogger to a computer via USB.

### 2.3. Calibration Theory and Process

The two chest motion waveforms, measured at the thorax (TC) and the abdomen (AB), are related to the lung volume waveform as per the model (reproduced in Equation (1)) introduced by Konno and Mead [21]. The lung volume waveform calculated using this model is an estimate of the actual lung volume and is denoted by $\hat{V}$. The parameters in curly braces denote vectors with a length of N elements. The Volume–Motion (V-M) parameters, Ktc and Kab, convert the TC and AB waveforms from arbitrary units (i.e., counts) to volume units (i.e., (mL)). These V-M parameters were obtained via calibration of the WRM, which must be done on the person wearing the WRM to account for variations in physiology and breathing style across participants and variations in respiratory sensor response across shirts. Although calibration is not necessary for deriving the inhalation duration, exhalation duration, and breath-hold, since these can be obtained directly from the chest motion waveform, it is necessary for deriving the inhalation and exhalation volumes and flow rates.
(1)$$\left\{\hat{V}\right\}=Ktc\times \left\{TC\right\}+Kab\times \left\{AB\right\}$$

During calibration, participants were instructed to breathe exclusively through a spirometer (Vernier, Beaverton, OR, USA) while wearing the WRM and a nose clip and perform the following breathing exercise while seated: 5 normal breaths, followed by a deep inhale, a breath-hold, then an exhale, followed by 5 more normal breaths. This was repeated 3 times without pausing in between. Participants were instructed to abort the breathing exercise at any time if they experienced any discomfort.

The spirometer measures the actual lung volume waveform (SP) and is considered the true condition. By using Equation (1), the $\hat{V}$ waveform can be calculated from the TC and AB waveforms and an arbitrary set of Ktc and Kab. The difference between $\hat{V}$ and SP is defined as the residual (r, Equation (2)).
(2)$$\left\{r\right\}=\left\{SP\right\}-\left\{\hat{V}\right\}$$

The residual is calculated at each value of the SP, TC, and AB waveforms and is therefore itself a waveform with a length of N elements. For a given set of SP, TC, and AB waveforms, r is a function of Ktc and Kab. The objective of the calibration is to find a pair of Ktc and Kab that minimizes r. For this, the mean absolute residual ($\tilde{r},$ Equation (3)) is considered and is plotted as a function of Ktc and Kab to form a mean absolute residual surface. The $\tilde{r}$ is a scalar with units of [mL] and represents the mean of the absolute difference between the volume estimate and the true condition.
(3)$$\tilde{r}=\frac{1}{N}\sum_{i=1}^{N}\sqrt{r_{i}{}^{2}}$$

### 2.4. Obtaining V-M Parameters from Multiple Calibrations

Section 2.3 described the theory and process for obtaining the V-M parameters from a single calibration. As per the study protocol, each participant performed 3 calibrations, one on each day of observation. As a result, there were 3 mean absolute residual surfaces for each participant. To find the set of Ktc and Kab that yields comparable $\tilde{r}$ when applied to all 3 calibrations, we proposed finding the centroid to the polygon formed by connecting the minimum point of each surface. The values of Ktc and Kab at the centroid were then used to estimate the lung volume waveform (Equation (1)) from TC and AB obtained while the participant used their tobacco product in the natural environment. This method of obtaining the values of Ktc and Kab is broadly applicable for N_cal_ ≥ 3 calibrations.

### 2.5. Respiration Topography Analysis

The data analysis was conducted using an added feature in RIT’s previously described Topography Analysis Program (TAP) [22]. First, the TC and AB time-series waveforms collected from the WRM were segmented into data for calibration, obtained during the calibration process, and data for calculating respiration topography, obtained during the natural environment observation. Both TC and AB waveforms were smoothed using a Savitzky–Golay 128-sample rolling-average filter. TAP then identified the minima in each waveform and then performed baseline compensation by first generating the baseline trend waveform for both TC and AB by piecewise interpolation between the minima on each waveform. The baseline trend waveform of each chest motion waveform was then subtracted from the original waveforms. The resultant baseline compensated waveform would then nominally have minima at 0. The algorithm then identified the maxima locations in each of the baseline compensated waveforms. The Ktc and Kab parameters were then obtained from the calibration data segment using the method described above. These parameters were then applied to Equation (1) along with the TC and AB from the natural environment data segment to obtain the volume estimate waveform.

The volume waveform was then segmented into individual breathing cycles by using the previously found extrema locations. The minimum at the start of a rising wave is the start of an inhale (and the start of the cycle), and the minimum at the end of a falling wave is the end of an exhale (and the end of the cycle). By considering the maxima, each breathing cycle could then be further segmented into an inhale, breath-hold, and exhale. Between any two minima there is one maximum, which corresponds to the end of the inhale. The segment of the waveform from the preceding minimum to this maximum was considered the inhale portion of the cycle. Subsequently, the period from the end of the inhale to the following minima was considered the exhale portion of the cycle. If a period of stable volume (neither rising nor falling significantly) is present following the end of an inhale before the start of the exhale, then this is considered the breath-hold period. During normal breathing, the breath-hold period is typically negligible; however, breath-holding is likely present following smoke/vapor inhalation during tobacco use.

### 2.6. Assessing Compliance and Acceptability

A daily paper study log was given to each participant that allowed them to self-report their product use during the study. The participant was instructed to report the number of cigarettes or hookah sessions they had each day along with the number of times they used their assigned product without wearing the WRM. By using this information, the participant’s compliance was assessed. The study log also allowed the participant to track whether they charged the WRM daily, which is necessary to prevent loss of data from the WRM discharging fully the following day.

During the exit appointment, each participant was given a questionnaire that assessed the acceptability of the study and the WRM. The questionnaire was administered as an interview, with the research administrator asking the questions and noting down the verbal responses from the participant.

## 3. Results

### 3.1. Study Cohort

A total of 47 individuals started the pre-screening survey (Figure 1), of whom 19 did not complete the survey. Of the 28 individuals who completed the survey, 15 were found to be ineligible. Reasons for ineligibility included not being above the legal smoking age or not being a cigarette or hookah user. Of the 13 eligible individuals, 10 responded to the follow-up email and scheduled an appointment. All 10 individuals were confirmed to be eligible during the intake appointment and were enrolled. All 10 enrolled participants were RIT students between the age of 21 and 29, of whom five were male, and five were female. Of these, three did not complete the study protocol: two were voluntary dropouts, and one was dismissed because they were non-compliant. Data from these three participants were not included in the analysis. Of the remaining seven participants, four completed the study as hookah users, two as cigarette users, and one (PS6-02) completed the study twice, once as a hookah user and once as a cigarette user. As such, there were eight sets of data (N_H_ = 5 Hookah and N_C_ = 3 Cigarette) presented for the waveform results and respiratory topography results. Enrolled participants’ responses to the pre-screening survey are shown in Table 2.

### 3.2. Waveform Results

All eight data sets from seven participants were analyzed. Each data set included three calibration sessions and at least one session of tobacco use observation in a natural environment. All hookah participants only had one tobacco use session, but all cigarette participants had more than one. A session is defined as the time from when the Hexoskin datalogger is plugged into the shirt until it is detached. The Hexoskin datalogger only collects data while it is attached to the shirt and creates a new data file every time it is connected to the shirt. This section presents the results of the analysis from the calibration process through to obtaining the volume estimate waveform using one participant’s data (PS6-11) as an exemplar case (Figure 2). Similar results were obtained across all data sets.

Figure 2a shows an exemplar calibration from participant PS6-11. The top panel in Figure 2a is the spirometer volume waveform (mL). The bottom two panels in Figure 2a are the corresponding uncalibrated TC and AB waveforms (counts), respectively. The spirometer volume was calculated as the integral of the measured flow rate.

Figure 2b shows the mean absolute residual surface generated from the SP, TC, and AB waveforms from Figure 2a, with Ktc and Kab each ranging from −50 to +50. The mean absolute residual surface shows that $\tilde{r}$ was more sensitive to Ktc than Kab and that there was a region of minima represented by the valley in the 3D surface. The black line represents the line of fit to the minima region; effectively, the 2D minima line to the 3D surface. The pair of Ktc and Kab that minimizes the surface is found by finding the location of the smallest value of $\tilde{r}$ along this line. For the set of data shown in Figure 2, the values of Ktc and Kab corresponding to the minima of the surface were found to be 7.17 and 13.84, respectively.

Figure 2c shows the minima lines from the three calibrations conducted with participant PS6-11. The circular markers indicate the global minimum point on each line of local minima. The x marker indicates the location of the centroid of the polygon formed from the circular markers at the vertices. For the set of calibrations shown in Figure 2, the values of Ktc and Kab at the centroid were found to be 14.67 and 13.52, respectively.

Exemplar TC and AB waveforms for participant PS6-11 are shown in Figure 3 for a single minute of one hookah session captured in the natural environment. As can be seen in the figure, the $\hat{V\;}$ waveform captured a mix of normal breaths (volumes of around 500 mL), deep breaths (volumes of around 1000 mL and greater), and breath-holds.

### 3.3. Respiration Topography

The computed respiration topography data of the five hookah data sets and three cigarette data sets are reported in Table 3. A total of 4733 breathing cycles were observed across all participants, of which 232 were from cigarette smokers, and 4501 were from hookah smokers. The number of breathing cycles from cigarette smokers made up around 5% of the total data. The low number of breathing cycles observed from cigarette smokers compared to hookah smokers is a direct consequence of the study protocol that only required participants to wear the shirts during tobacco use. Since cigarette sessions are in the order of minutes and a hookah session is in an order of hours, the total breaths observed from cigarette users were much less than that from hookah smokers. There were also more hookah data sets than there are cigarette data sets.

Figure 4 shows exemplar descriptive statistics collected for participant PS6-04. The histograms in Figure 4 show three characteristics: (1) The inhalation and exhalation volume distributions (Figure 4a) were overlapping, indicating that there was the conservation of mass across the spectrum of breaths observed, which was in contrast to (2) The duration (Figure 4b) and the flow rate distributions (Figure 4c) both showed a distinct but consistent bias between inhalation and exhalation, with inhales being shorter and sharper and exhales being longer and more relaxed. (3) The volume distribution (Figure 4a) was right-skewed, with the mode (of around 200 mL) smaller than the mean (of around 300 mL), which was indicative of a higher frequency of smaller volume breaths than deeper breaths. The duration and the flow rate distributions also appeared to be right skewed, but this behavior was not seen consistently across all participants.

The grand mean of the ratio between the inhale volume to the corresponding exhale volume within a breath across all breaths captured across all participants was 1.04. A volume ratio close to 1 is indicative of the mass conservation on a breath-by-breath basis. This is visually illustrated by the scatter points clustering around the 1-to-1 line in the scatter plot of inhalation versus exhalation volume in Figure 4d. The inhale volume should be the same as the exhale volume to satisfy the conservation of mass. However, there is some potential for variability in small time windows, e.g., on a breath-by-breath basis, due to the variability in breathing behavior and the ability for a person to tap into their lung’s reserve capacity. This may explain some of the scatter points that lie farther away from the 1-to-1 line. It is expected that the conservation of mass should hold over time, which can be assessed by comparing the cumulative inhale volume to the cumulative exhale volume. The ratio of cumulative inhale volume to the cumulative exhale volume, summed across all participants, was found to be 0.996.

The grand mean of the ratio between the inhale duration to the corresponding exhale duration within a breath across all breaths captured across all participants was 0.654. This ratio being smaller than 1 is indicative of the bias towards the exhale being longer than the inhale. This is visually illustrated by the larger amount of scatter points above the 1-to-1 line in Figure 4e.

The grand mean of the ratio between the inhale flow rate to the corresponding exhale flow rate within a breath across all breaths captured across all participants was 1.32. This ratio being larger than 1 is indicative of the bias towards sharper inhales than exhales. Since the flow rate was calculated from the ratio of the volume to the duration, this flow rate bias was inversely related to the duration bias. This is visually illustrated by the larger amount of scatter points below the 1-to-1 line in Figure 4f.

The consistency between inhale and exhale volumes and the bias between inhale and exhale duration and flow rates were likewise observed in all participants (Figure 5).

Figure 5a shows the mean and its 95% confidence interval of inhale and exhale volumes across all participants. The confidence intervals were larger in the cigarette cases than the hookah cases due to the smaller number of breathing cycles observed in the cigarette cases.

Figure 5b shows the mean and its 95% confidence interval of inhale, exhale, and cycle durations across all participants. In all participants, the mean exhale duration was longer than the mean inhale duration. In the absence of breath-holding, the cycle duration was the sum of the inhale and exhale duration.

Figure 5c shows the mean and its 95% confidence interval of inhale and exhale flow rates for all participants. The mean inhale flow rate was higher than the mean exhale flow rate for every participant. This flow rate bias was consistent with the opposite bias in inhale and exhale durations.

### 3.4. Compliance and Acceptance of the Monitor

All seven enrolled participants took the exit questionnaire and indicated that the Hexoskin shirt did not change their typical tobacco use behavior. However, one participant indicated that wearing the shirt made them feel like they were being monitored. Two participants indicated the shirt was “tight”, while five indicated it was either “comfortable” or “fine”. All participants reported wearing the shirt for at least an hour, with two reporting wearing it for 4–5 h, and one reported wearing it for the whole day. All but one participant indicated that they wore the shirt only for the duration of smoking. All participants connected the datalogger only while they were smoking. All participants indicated that they felt comfortable wearing the shirt in a public setting. Four participants indicated the requirement to wear the shirt had no impact on their desire to smoke, but the other three indicated it was either slightly discouraging or hard to remember to put on. All but one participant indicated that the shirt had no effect on their respiratory effort to puff their tobacco product. One participant indicated that the shirt had a minor impact on their ability to perform their regular non-smoking activities. When asked for the maximum amount of time they would be willing to wear the Hexoskin shirt, two participants answered that they would be willing to wear it for less than 3 h, two participants answered up to 7 h, and three said they would be willing to wear it for a day or more.

## 4. Discussion

Many calibration techniques have been introduced in the literature [21,23,24,25]. We extended upon previous methods by analyzing the sensitivity of the residual to each V-M parameter. We saw that the mean absolute residual surface (Figure 2b) showed a significantly stronger sensitivity to Ktc than to Kab. This feature was present in all participants. A simple linear regression algorithm would have just returned a single pair of Ktc and Kab that was the “good fit”, but we see that there is potentially an infinite number of combinations of Ktc and Kab with residuals within close proximity of each other (as illustrated by the valley in the mean absolute residual surface in Figure 2b). In the absence of more information, it was not possible to determine the pair of Ktc and Kab that would qualify as a “good fit” beyond just a value judgment on the part of the analyst. One potential way to facilitate this is by obtaining more information about the physiology and the breathing style of the participant, such as by calibrating under different body postures or breathing maneuvers and patterns. We believe that the method presented here is a reasonable first step towards a more robust way to derive Ktc and Kab than the linear regression method presented in the past.

Additionally, we prototyped the logic to derive a single pair of Ktc and Kab from multiple calibrations (N_cal_ = 3 in this paper) using the centroid method. Other approaches considered included: (i) taking the mean Ktc and the mean Kab of the N_cal_ calibrations, (ii) only use a pair of Ktc and Kab from one of the N_cal_ calibrations, (iii) applying the pair of Ktc and Kab from the most recent calibration for the data collected following the calibration leading up to the next calibration, (iv) Combining all N_cal_ calibrations into a single data set to determine a single pair of Ktc and Kab. It is presently unclear what the best method is from these, but we plan on investigating this further.

The calibrations were conducted while the participants were seated only, with their torso perpendicular to the ground, which we deemed to be the posture the participants would most likely use during their tobacco use sessions. Some prior research suggests that a participant’s V-M parameters may vary based on their body position [26]. Hence, it is possible that the inhalation and exhalation volumes measured by the WRM calibrated while seated may not be accurate if the participants were not seated while smoking. In a future study, we plan to investigate the impact of calibration posture on the accuracy of the volume measurements. The duration-based respiration topography parameters, such as inhale and exhale duration, were not impacted by the magnitude of the volume waveform and, therefore not impacted by the calibration since they were calculated from the temporal locations of the extrema on the lung volume waveform.

We were able to deploy the monitor for a 3-day observation study, which covered at least one full 24-h period. We were also able to analyze all the data collected from this period of observation and obtain both lung volume waveform and respiration topography for all seven participants. For this, we developed a computer algorithm to: (i) pre-process the data from the Hexoskin, (ii) perform the calibration and apply it to the measured TC and AB waveforms to obtain the lung volume waveform, and (iii) derive the respiration topography. This was necessary because the Hexoskin did not come with the necessary software to obtain either lung volume or respiration topography [1]. We did not modify the Hexoskin hardware in any way for this study.

The data presented here were from participant’s smoking sessions and comprised two predominant underlying behaviors: natural tidal breathing and smoking-related ventilation, i.e., inhalation of tobacco emissions post puffing with potential breath-holding prior to exhalation. Although we did not attempt to discriminate between the two behaviors in this paper, the distribution (Figure 4) of larger amounts of sub-1000 mL volumes and smaller amounts of volumes greater than 1000 mL is indicative of natural tidal breaths and smoking-related ventilation, respectively. The ability to discriminate between the two behaviors is necessary for identifying smoking-specific respiration topography. This is currently a focus of our future development. For this, we will leverage our wPUM topography monitors [27] to identify the start and end of a puff relative to the lung volume waveform measured by the WRM. This will help in discriminating the breaths that are associated with smoking from the natural tidal breaths. Additionally, this will potentially allow us to discriminate between MTL and DTL. The characterization of these smoking patterns is also a topic for future research.

It was observed that in all participants, the mean exhale duration was longer than the mean inhale duration. This is consistent with the accepted pattern of tidal breathing in healthy adults. We predict that this phenomenon is also present in smoking-related ventilation cycles as a mechanism for increasing the nicotine uptake by extending the smoke retention time. Additionally, changes in the duration of either the inhale or the exhale or the ratio between the two may also be an effect of compensatory behavior. As expected from the relationship between the exhale duration and inhale duration and the consistent volume inhaled and exhaled, the data presented here indicate that the mean inhale flow rate was higher than the mean exhale flow rate.

The extrema locations in the volume waveform had a strong impact on the respiration topography since every respiration topography parameter was calculated based on the location of the extrema that prescribes each breathing cycle. Therefore, the algorithm used to find these extrema points had an impact since any misplaced extrema will affect the respiration topography of that breathing cycle and possibly those of the adjacent breathing cycles. The quality of the data and the nature of the behavior being observed also has an impact on the algorithm’s ability to obtain respiration topography, including irregularities in the volume waveform that may be a result of behaviors, such as walking, talking, and yawning, or sensor slippage on the body. More work still needs to be done to assess the sensitivity of respiration topography to these irregularities in the volume waveform. The current algorithm operated on the volume waveform directly and used the prominence values of the peaks and troughs of the breathing cycles to find the extrema, but alternative approaches exist that make use of frequency domain analysis or machine learning. More work must be done and presented to make a meaningful comparison between methods of obtaining respiration topography from lung waveform.

The participants’ acceptability of the Hexoskin is key to assessing its viability as a tool for ambulatory measurement of respiration topography. Prior to the start of the study, we were concerned that the participants would not be willing to wear the Hexoskin shirt for extended periods, i.e., to cover the whole day of observation. Therefore, we only required the participant to wear the shirt while they were using their tobacco product, and we expected that the participants would take the shirt off in between sessions. From the exit interview results, we can see that the majority did remove the shirt in between sessions but most indicated that they would be willing to wear it for extended periods. In a future study, we intend to recommend the participant wear the shirt for a full day of observation. Data collected outside of their smoking session will help establish the participant’s baseline tidal breathing behavior and help with identifying smoking-related ventilation. The downside to the requirement to wearing the shirt for an extended amount of time is that we would lose the natural indication of the start and end of a smoking session that came with the participant attaching and detaching the datalogger from the shirt. Another option is for the participant to wear the shirt throughout the day but detach the datalogger between sessions. The disadvantage here is that there is a risk that the participant would forget to reattach the datalogger prior to starting a session, causing the loss of valuable data. Alternatively, we can leverage our wPUM topography monitor to provide the start and end time of each puff and the start and end time of each session.

Another concern that we had was in the amount of data that would be collected if the shirt was worn for an extended time. The Hexoskin samples each chest motion sensor at 128 Hz, which after 12 h would result in over 5 million samples. In addition to the storage concerns, we anticipated computational difficulty with processing large vectors and finding extrema locations in large sets of data. In practice thus far, we have not encountered any issues with our algorithm in terms of the data size. We expect that the algorithm would be scalable to larger datasets obtained from longer observation periods.

## 5. Conclusions

A method for calibrating a commercially WRM across multiple observation days was presented. The thoracic and the abdominal signals were measured using the Hexoskin Smart Garment on seven participants while they were using their tobacco product. Lung volume waveforms and respiration topography parameters were obtained using an extension of RIT’s Topography Analysis Program. The acceptability of the Hexoskin was assessed, with most participants indicating a willingness to wear the shirt for an extended amount of time, allowing for longer periods of observation. This is the first study to demonstrate ambulatory measurement of lung volume in the natural environment. The ability to quantify respiration topography, combined with existing techniques of capturing natural environment puffing topography, will lead to a comprehensive, objective assessment of tobacco use behavior.

## Figures and Tables

**Figure 1 ijerph-18-07012-f001:**
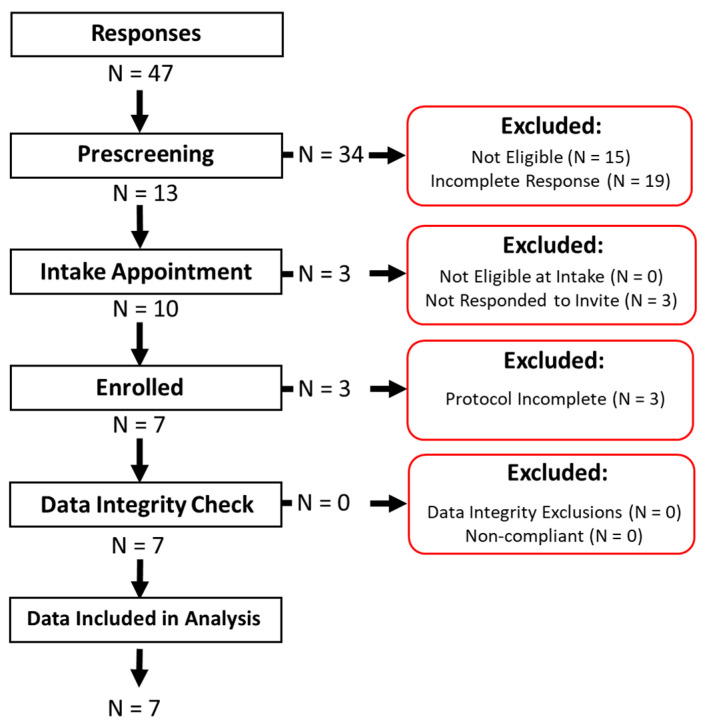
Participant recruitment, screening, and enrolment flow chart.

**Figure 2 ijerph-18-07012-f002:**
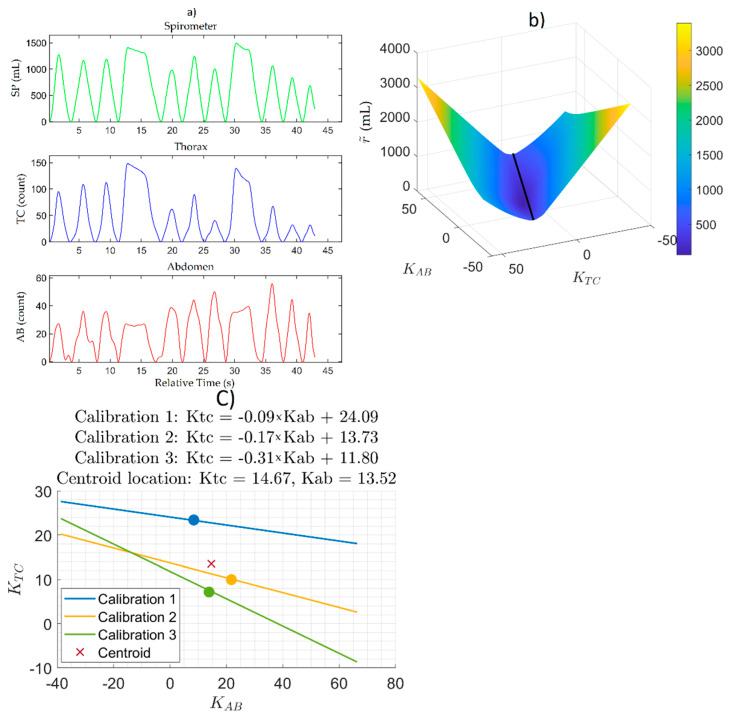
(**a**) SP, TC, and AB waveforms from an exemplar calibration conducted with a participant; (**b**) The mean absolute residual surface generated from the data in (**a**) as a function of Ktc and Kab. The black line indicates the line of minima; (**c**) The lines of minima from the mean absolute residual surfaces from the three calibrations conducted with the participant, the minima point on each line (circle), and the calculated centroid minimum (x).

**Figure 3 ijerph-18-07012-f003:**
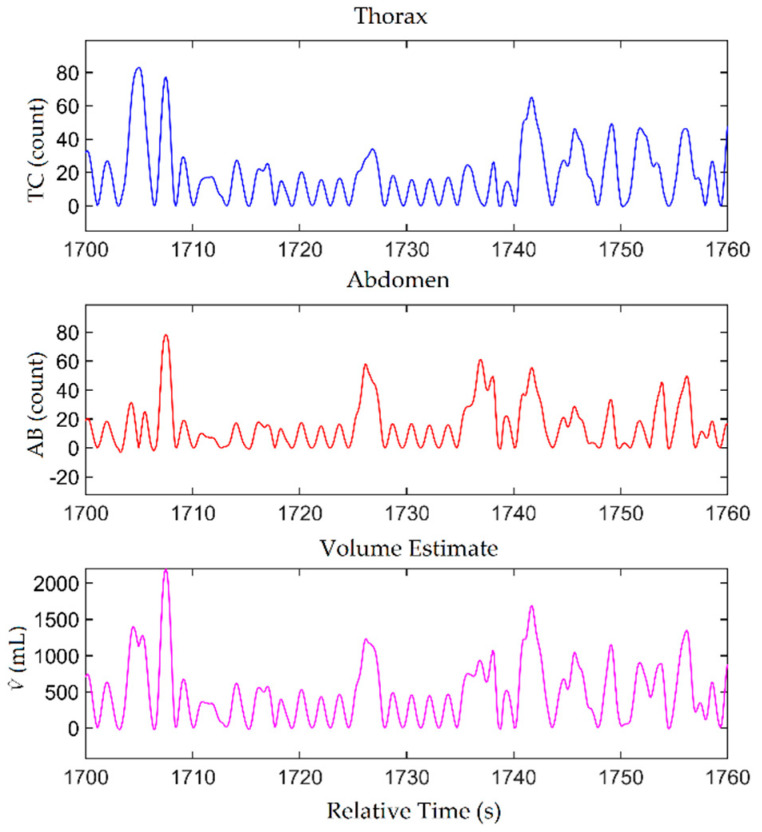
Exemplar uncalibrated TC and AB waveforms from a 1-minute subsection of a hookah session observed in the natural environment. The $\hat{V}$ waveform was calculated from the TC and AB waveforms using the Ktc and Kab values obtained using the centroid method.

**Figure 4 ijerph-18-07012-f004:**
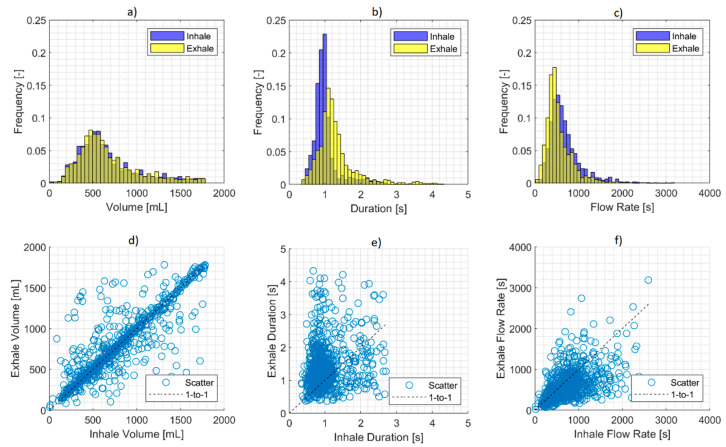
Exemplar respiration topography results from a participant (PS6-04, 1229 breathing cycles observed). Top row—Distribution of inhalation and exhalation: (**a**) volumes; (**b**) duration; (**c**) flow rates. Bottom row—Breath-by-breath comparison of inhalation to exhalation: (**d**) volumes; (**e**) durations; (**f**) flow rates.

**Figure 5 ijerph-18-07012-f005:**
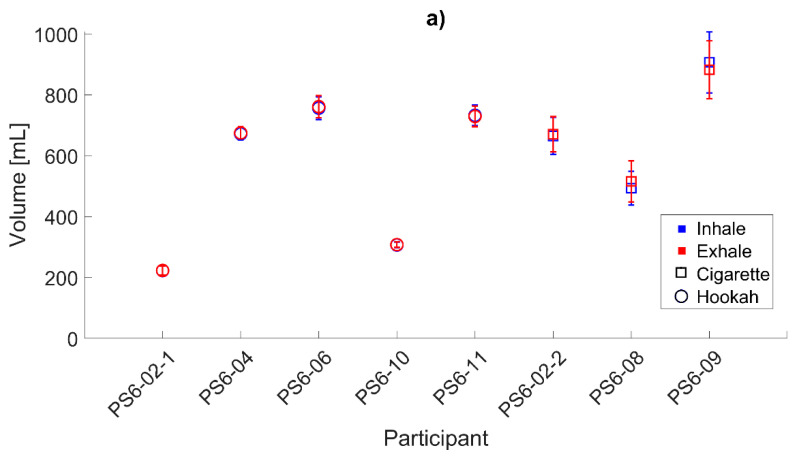
The mean and its 95% confidence interval (**a**) inhale and exhale volumes; (**b**) inhale, exhale, and cycle durations; (**c**) inhale and exhale flow rates across all participants. The mean was calculated from all observed breaths across all sessions and days by that participant. Blue, red, and black colors denote inhale, exhale, and cycle, respectively. Circular markers indicate hookah participants, and square markers indicate cigarette participants.

**Table 1 ijerph-18-07012-t001:** Questions presented to the participant on the pre-screening survey. The answer to Question 1 was used as part of the Inclusion Criteria.

	Question	Answer Type
Q1	In the last 30 days how often did you use a cigarette?	Multiple choice: “Everyday”, “Some days”, “Not at all”
Q2	In the last 30 days how often did you use a hookah?	Multiple choice: “Everyday”, “Some days”, “Not at all”
Q3 *	In the last 30 days, how many days did you use a cigarette?	Text entry
Q4 *	In a typical week, how many days do you use a cigarette?	Text entry
Q5 *	How long ago did you start using a cigarette?	Text entry
Q6 ^	In the last 30 days, how many days did you use a hookah?	Text entry
Q7 ^	In a typical week, how many days do you use a hookah?	Text entry
Q8 ^	How long ago did you start using a hookah?	Text entry

* Question only presented if participant answered “Everyday” or “Some days” in Q1; ^ Question only presented if participant answered “Everyday” or “Some days” in Q2.

**Table 2 ijerph-18-07012-t002:** Enrolled participants’ responses to the pre-screening survey (Table 1). N/A = Not Applicable, indicating questions that were not asked because of the responses to Q1 and Q2.

Participant	Gender	Age	Q1—Cigarette Use	Q2—Hookah Use	Q3—Days used Cigarette in Last 30 Days	Q4—Days Used Cigarette in Typical Week	Q5—How Long since Started Cigarette Use	Q6—Days Used Hookah in Last 30 Days	Q7—Days Used Hookah in Typical Week	Q8—How Long since Started Hookah Use
PS6-02	M	27	Some days	Some days	2	1	3 Years	6	2	5 Years
PS6-04	M	23	Some days	Some days	5	1	1 Year	5	2	1 Month
PS6-06	F	29	Some days	Some days	1	1	1 Day	23	7	10 Days
PS6-08	F	22	Everyday	Not at all	30	7	3 Years	N/A	N/A	N/A
PS6-09	F	25	Everyday	Not at all	30	7	6 Years	N/A	N/A	N/A
PS6-10	F	22	Not at all	Some days	N/A	N/A	N/A	7	1	4 Years
PS6-11	M	23	Some days	Some days	15	1	10 Years	9	1	4 Years

**Table 3 ijerph-18-07012-t003:** Mean respiration topography summary by participant.

Participant	Product Type	Cycle Count	Mean Inhale Volume (STD) (mL)	Mean Exhale Volume (STD) (mL)	Mean Inhale Flow Rate (STD) (mL/s)	Mean Exhale Flow Rate (STD) (mL/s)	Mean Inhale Duration (STD) (s)	Mean Exhale Duration (STD) (s)	Mean Cycle Duration (STD) (s)
PS6-02-1	Hookah	405	223 (157)	223 (155)	194 (137)	140 (121)	1.24 (0.54)	1.88 (0.93)	3.12 (1.21)
PS6-04	Hookah	1229	672 (358)	675 (356)	690 (371)	558 (328)	1.01 (0.37)	1.32 (0.59)	2.33 (0.77)
PS6-06	Hookah	381	756 (373)	761 (362)	775 (476)	607 (391)	1.11 (0.5)	1.54 (0.89)	2.65 (1.11)
PS6-10	Hookah	2009	307 (213)	308 (213)	270 (180)	210 (168)	1.21 (0.46)	1.76 (0.81)	2.98 (1.03)
PS6-11	Hookah	477	733 (375)	729 (376)	739 (399)	606 (400)	1.06 (0.45)	1.44 (0.82)	2.5 (1.04)
PS6-02-2	Cigarette	109	665 (319)	670 (304)	657 (345)	526 (306)	1.13 (0.49)	1.46 (0.69)	2.59 (1)
PS6-08	Cigarette	61	493 (218)	516 (265)	496 (207)	269 (199)	1.01 (0.26)	2.4 (1.08)	3.41 (1.13)
PS6-09	Cigarette	62	907 (397)	883 (374)	1020 (731)	752 (652)	1.13 (0.57)	1.81 (1.08)	2.94 (1.26)
	Hookah Group Mean	N_H_ = 5	538 (253)	539 (254)	534 (278)	424 (229)	1.13 (0.1)	1.59 (0.23)	2.72 (0.33)
	Cigarette Group Mean	N_C_ = 3	689 (208)	690 (184)	724 (269)	516 (242)	1.09 (0.07)	1.89 (0.47)	2.98 (0.41)
	Grand Mean	4733	492 (358)	494 (357)	482 (388)	381 (339)	1.13 (0.46)	1.61 (0.82)	2.74 (1.04)

## Data Availability

The data presented are contained within the article.

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
