# Peer review of "Feasibility of Using the Hexoskin Smart Garment for Natural Environment Observation of Respiration Topography"

_ijerph, 2021, doi:10.3390/ijerph18137012_

Round 1
Reviewer 1 Report
- The presented work is a primitive study. It is limited to the optimisation problem of the Ktc and Kab parameter in the already existing model introduced by Konno and Mead. The author discussed the results based on inhale and exhale duration only. These results, e.g. inhale and Exhale duration, can also be inferred from the hexoskin data directly. The significance of derived parameter volume is not discussed explicitly in the analysis.
- Also, Without a reference device, Hexoskin can not be used for the proposed feasibility study. The Hexoskin shirt needs to be calibrated every time with a reference device for each participant.
- More emphasis is given to future work. How the present work can be corroborated with the proposed future work is missing.
- Line number 115: “The feasibility of using the Hexoskin for ambulatory measurement of respiration topography was previously assessed.” The author should provide a reference for the statement.
- The conducted experiments are not monitored. There should be some monitored experiments to validate the unmonitored duration data.
- Fig 4 (c) and (f) needs correction. The x-axis and y-axis representation are incorrect.
- What does N refer to in Table 3? Either use the same nomenclature in the text while explaining the fields of the Table or modify the table fields. At line 341, N represents the number of participants, while at line 375, N represents the number of calibrations.
- The grammar and spacing errors can be improved.
Reviewer 2 Report
Dear authors,
the manuscript presents a method that can create a benefit for the scientific community in this domain.
However,
Major revision:
Line 115- Missing ref
From line 113 there is not a description of the smart cloth and a photo. A depth description should be there to clarify how is possible to gather these data.
Please check the missing unit of measurements in the entire paper.
Did you compare these results with a gold standard RM?
It is the population valid as only N=5 Hookah and N=3 Cigarette participants were enrolled.
I propose to extend this experiment to a broad population as you consider this as a method of calibration.
All the manuscript is not well balanced in the description. Please avoid that.
Reviewer 3 Report
The topic "Feasibility of Using the Hexoskin Smart Garment for Natural Environment Observation of Respiration Topography" is interesting. However, some minor concerns below maybe needed more addressed before considering to be published.
1.One of the purposes in this study is to calibrate the data from the Hexoskin Smart Garment. It would be better to describe more what the biases of the signal would be induced form the product ? is there any process from the original design?
2.In page 1, authors describe "smokers were recruited for a 3-day natural environment observation study". What kind of natural environment and the activity while acquiring the data from the participants?
And how to decide which section of typical signal can be extract to analyze?
3.Figure 2.a shows an exemplar calibration from participant PS6-11.The unit of horizontal axis is "time "in second and it showed the duration is 50 seconds. Is it enough for data analysis? How it could be defined?
4.Figure 2.c shows the minima lines from the 3 calibrations conducted with participant 250 PS6-11. Calibration 1: Ktc=-0.09*Kab+24.09; Calibration 2: Ktc=-0.17*Kab+13.73;Calibration 3: Ktc=-0.31*Kab+11.80; When Kab=0, the y intercept (Ktc) would be 24.09, 13.73 and 11.8, respectively. However, it showed that y intercept of Calibration 1 is 28, Calibration 2 is 20, Calibration 3 is 24. Something wrong with the calculation or the figure?
5.Each data would have three calibration functions with different parameters. Which one could be decide to adopted to calibrate the data? Moreover, how the calibration function could be defined and applied for each participant?
6.Figure 3. indicated exemplar uncalibrated TC and AB wave forms from a 1-minute subsection of a hookah session observed in the natural environment. How the relative time "1700"~"1760" was defined?
7.Figure 4. revealed exemplar respiration topography results from a participant. But figure (c) and (f) were left and right inverse ?
8.If there are data from non-smokers that can compare with the smokers?
9.Can the authors describe more applications of chest motion waveform and respiration topography by Hexoskin Smart Garment?
10.If it is possible, please add recent references in 3 years more.
Round 2
Reviewer 1 Report
I am happy with the author's response.